# Inducing Kondo screening of vacancy magnetic moments in graphene with gating and local curvature

Yuhang Jiang[1], Po-Wei Lo[2,3,7], Daniel May[4], Guohong Li[1], Guang-Yu Guo[2,3], Frithjof B. Anders[4], Takashi Taniguchi[5], Kenji Watanabe[5], Jinhai Mao[1,6] & Eva Y. Andrei[1]

In normal metals the magnetic moment of impurity-spins disappears below a characteristic Kondo temperature which marks the formation of a cloud of conduction-band electrons that screen the local-moment. In contrast, moments embedded in insulators remain unscreened at all temperatures. What then is the fate of magnetic-moments in intermediate, pseudogap systems, such as graphene? Theory predicts that coupling to the conduction-band electrons will drive a quantum phase transition between a local-moment phase and a Kondo-screened phase. However, attempts to experimentally confirm this prediction and its intriguing consequences, such as electrostatically tunable magnetic-moments, have been elusive. Here we report the observation of Kondo-screening and the quantum phase-transition between screened and unscreened phases of vacancy magnetic moments in graphene. Using scanning tunneling spectroscopy and numerical renormalization-group calculations we show that this transition enables to control the screening of local moments by tuning the gate voltage and the local curvature of the graphene membrane.

[1] Department of Physics and Astronomy, Rutgers University, 136 Frelinghuysen Road, Piscataway, NJ 08855, USA. [2] Department of Physics, National Taiwan University, Taipei 10617, Taiwan. [3] Physics Division, National Center for Theoretical Sciences, Hsinchu 30013, Taiwan. [4] Theoretische Physik 2, Technische Universität Dortmund, 44221 Dortmund, Germany. [5] Advanced Materials Laboratory, National Institute for Materials Science, 1-1 Namiki, Tsukuba 305-0044, Japan. [6] Institute of Physics & University of Chinese Academy of Sciences, Chinese Academy of Sciences, Beijing 100190, China. [7] Present address: Department of Physics, Cornell University, Ithaca, NY 14853, USA. Correspondence and requests for materials should be addressed to J.M. (email: jhmao@ucas.edu.cn) or to E.Y.A. (email: andrei@physics.rutgers.edu)

Graphene, with its linear density of states (DOS) and tunable chemical potential[1,2], provides a playground for exploring the physics of the magnetic quantum phase transition[3–9] (Fig. 1a). But embedding a magnetic moment and producing sufficiently large coupling with the itinerant electrons in graphene, poses significant experimental challenges: adatoms typically reside far above the graphene plane, while substitutional atoms tend to become delocalized and non-magnetic[10]. An alternative and efficient way to embed a magnetic moment in graphene is to create single atom vacancies. The removal of a carbon atom from the honeycomb lattice induces a magnetic moment stemming from the unpaired electrons at the vacancy site[11–14]. This moment has two contributions: one is a resonant state (zero mode-ZM) at the Dirac point (DP) due to the unpaired electron left by the removal of an electron from the π-band; the other arises from the broken σ-orbitals, two of which hybridize leaving a dangling bond that hosts an unpaired electron[14]. The ZM couples ferromagnetically to the dangling σ-orbital[14], as well as to the conduction electrons[15,16] and remains unscreened. In flat graphene the magnetic moment from the dangling σ-bond is similarly unscreened because the σ-orbital is orthogonal to the π-band conduction electrons[16,17]. However, it has been proposed that this constraint would be eliminated in the presence of a local curvature which removes the orthogonality of the σ-orbital with the conduction band, and enables Kondo screening[16,18,19]. One strategy to introduce local curvature is to deposit the flexible graphene membrane on a corrugated substrate.

Here we employ the spectroscopic signature of the Kondo effect to demonstrate that screening of vacancy magnetic moments in graphene is enabled by corrugated substrates.

Crucially, variations in the local curvature imposed by the corrugated substrate provide a range of coupling strengths, from subcritical to the supercritical regime, all in the same sample. An unexpected consequence of this unusually wide variation of coupling strengths is that global measurements such as magnetization[20,21] or resistivity[22] can give contradictory results. In fact, as we will show, the quantum critical transition between Kondo screened and local moment phases in this system, can only be observed through a local measurement.

## Results

**Scanning tunneling microscopy and spectroscopy.** We employed scanning tunneling spectroscopy (STS)[23,24] to identify Kondo screening of the vacancy magnetic moment by the distinctive zero-bias resonance it produces in the $dI/dV$ curves ($I$ is the tunneling current and $V$ the junction bias), hereafter called Kondo peak. We first discuss samples consisting of two stacked single layer graphene sheets on a $SiO_2$ substrate (G/G/$SiO_2$) capping a doped Si gate electrode (Fig. 1b). A large twist angle between the two layers ensures electronic decoupling, and preserves the electronic structure of single layer graphene while reducing substrate induced random potential fluctuations[24–26]. A further check of the Landau-level spectra in a magnetic field revealed the characteristic sequence expected for massless Dirac fermions[2,23], confirming the electronic decoupling of the two layers (Supplementary Note 1). Vacancies were created by low energy (100 eV) He$^+$ ion sputtering followed by in situ annealing[22,27,28]. In STM topography of a typical irradiated sample (Fig. 1c) the vacancies appear as small protrusions on top of large background corrugations. To establish the nature of a vacancy we

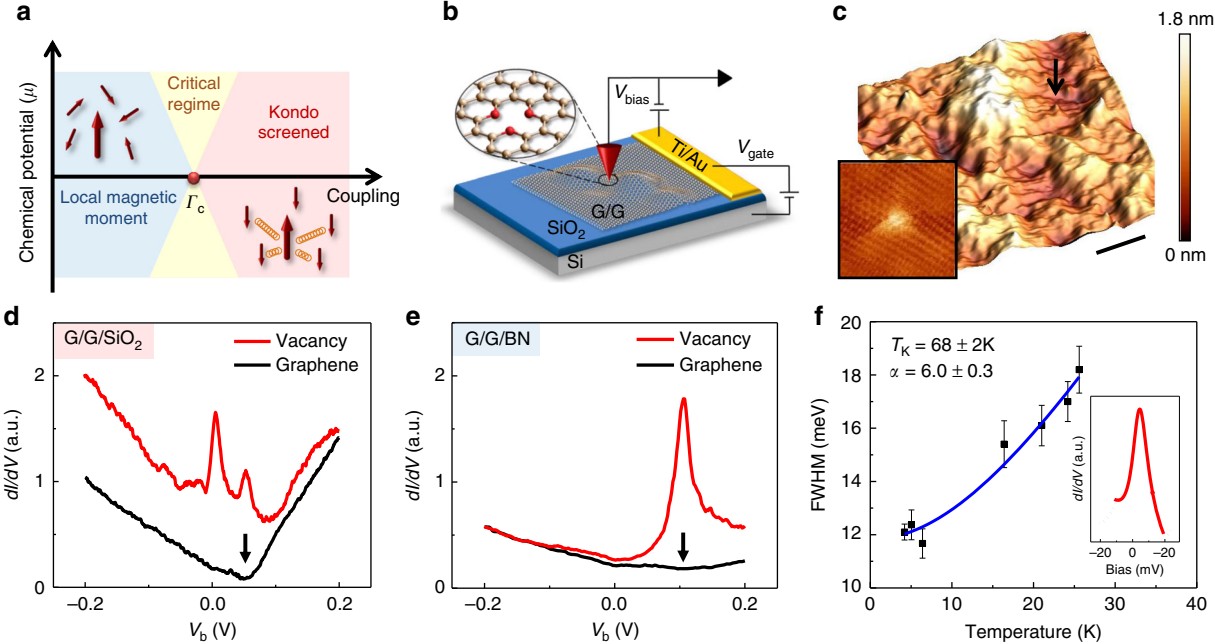

**Fig. 1** Kondo peak at a single-atom vacancy in graphene. **a** Schematic phase diagram of the pseudo-gap Kondo effect. The critical regime (yellow) separates the Local-magnetic-moment phase from the Kondo-screened phase. Arrows represent the ground state of the system with the large arrows corresponding to the local spin and the smaller ones representing the spins of electrons in the conduction band. **b** Schematics of the experimental setup. **c** STM topography of a double layer graphene on $SiO_2$ (G/G/$SiO_2$). The arrow indicates an isolated vacancy ($V_b = -300$mV, $I = 20$pA, $V_g = 50$ V). The scale bar is 20 nm. Inset: atomic resolution topography of a single atom vacancy shows the distinctive triangular structure (4 nm × 4 nm), $V_b = -200$mV, $I = 20$pA, $V_g = -27$V. **d** $dI/dV$ spectra at the center of a single atom vacancy (upper red curve) and on pristine graphene far from the vacancy (lower black curve). The curves are vertically displaced for clarity ($V_b = -200$mV, $I = 20$pA, $V_g = 0$ V). The arrow labels the Dirac point. **e** Same as **d** but for a vacancy in a G/G/BN sample ($V_b = -200$mV, $I = 20$pA, $V_g = -30$V). **f** Evolution of the measured full width at half maximum (FWHM) of the Kondo peak with temperature (black data points) shown together with the fit (blue solid line) discussed in the text. Error bars represent the linewidths uncertainty obtained from fitting the Kondo peak to a Fano lineshape. Inset: Zoom into the Kondo peak (black dotted line) together with the Fano lineshape fit (red solid line)

zoom in to obtain atomic resolution topography and spectro-scopy. Single atom vacancies are recognized by their distinctive triangular $\sqrt{3} \times \sqrt{3}\, R30°$ topographic fingerprint (Fig. 1c inset)[27–29] which is accompanied by a pronounced peak in the $dI/dV$ spectra at the DP reflecting the presence of the ZM. If both these features are present we identify the vacancy as a single atom vacancy (Supplementary Note 2) and proceed to study it further. In order to separate the physics at the DP and the Kondo screening which produces a peak near Fermi energy, $E_F \equiv 0$, the spectrum of the vacancy in Fig. 1d is taken at finite doping corresponding to a chemical potential, $\mu \equiv E_F - E_D = -54$ meV. Far from the vacancy (lower curve), we observe the V shaped spectrum characteristic of pristine graphene, with the minimum identifying the DP energy. In contrast, at the center of the vacancy (Fig. 1d upper curve), the spectrum features two peaks, one at the DP identifying the ZM and the other at zero bias coincides with the position of the expected Kondo peak[3]. (In STS the zero-bias is identified with $E_F$.) From the line shape of the zero-bias peak (Fig. 1f inset), we extract $T_K = (67 \pm 2)$K by fitting to the Fano line shape[30,31] characteristic of Kondo resonances (Supplementary Note 3). As a further independent check we compare in Fig. 1f the temperature dependence of the linewidth to that expected for a Kondo-screened impurity[30,32] (Supplementary Note 4), $\Gamma_{LW} = \sqrt{(\alpha k_B T)^2 + (2k_B T_K)^2}$ from which we obtain $T_K = (68 \pm 2)$ K, consistent with the above value, and $\alpha = 6.0 \pm 0.3$ in agreement with measurements and numerical simulations on ad-atoms[30,33]. Importantly, as we show below, this resonance is pinned to $E_F$ over the entire range of chemical potential values studied, as expected for the Kondo peak[34,35].

The gate dependence of the spectra corresponding to the hundreds of vacancies studied here falls into two clearly defined categories, which we label type I, and type II. In Fig. 2a we show the evolution with chemical potential of the spectra at the center of a type I vacancy. Deep in the p-doped regime, we observe a peak which is tightly pinned to, $E_F$, consistent with Kondo-screening. Upon approaching charge neutrality the Kondo peak disappears for $\mu \geq -58$ meV and reenters asymmetrically in the n-doped sector, for $\mu \geq 10$ meV. As we discuss below, the absence of screening close to the charge neutrality point and its reentrance in the n-doped regime for type I vacancies is indicative of pseudogap Kondo physics for subcritical coupling strengths[8,36]. For type II vacancies, the evolution of the spectra with chemical potential, shown in Fig. 2b, is qualitatively different. The Kondo peak is observed in the p-doped regime and disappears close to charge neutrality, but does not reappear on the n-doped side. We show below that this behavior is characteristic of pseudogap Kondo physics for vacancies whose coupling to the conduction band is supercritical[8,36].

**Numerical renormalization group calculations**. To better understand the experimental results we performed numerical-renormalization-group (NRG) calculations for a minimal model based on the pseudogap asymmetric Anderson impurity model (AIM)[6,16,37] comprising the free local σ-orbital coupled to the itinerant π-band (Supplementary Note 6). This model gives an accurate description of the experiment in the p-doped regime where the ZM is sufficiently far from the Kondo peak so that their overlap is negligible. Upon approaching charge neutrality, inter-actions between the two orbitals through Hund's coupling and level repulsion become relevant. As described in Supplementary Note 6 we introduced an effective Coulomb interaction term to take into account this additional repulsion. The single orbital model together with this phenomenological correction captures the main features of the Kondo physics reported here (Fig. 3).

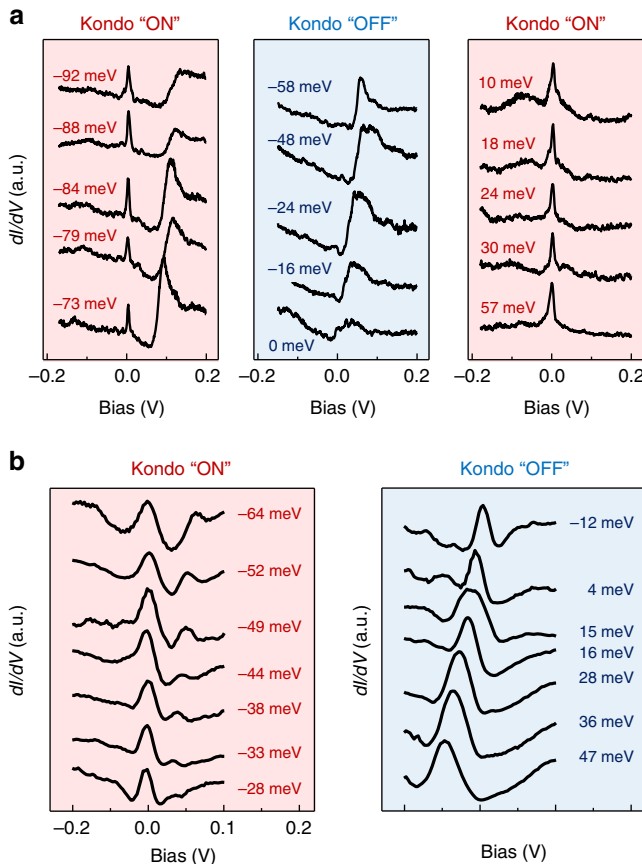

**Fig. 2** Evolution of Kondo screening with chemical potential. **a** $dI/dV$ curves for a subcritical Kondo vacancy (type I in text) with reduced coupling strength $\Gamma_0/\Gamma_C = 0.90$ at the indicated values of chemical potential. Red (blue) shade indicates the presence (absence) of the Kondo peak ($V_b = -200$mV, $I = 20$pA). The chemical potential is tuned by the backgate voltage[28]. **b** $dI/dV$ curves for a supercritical Kondo vacancy (type II in text) with $\Gamma_0/\Gamma_C = 1.83$

Results from a comprehensive NRG calculation using a two-orbital pseudogap AIM to model the problem[38] similarly indicate that this simplified one-orbital approach qualitatively describes the experimental results. The single orbital AIM is characterized by three energy scales, $\varepsilon_d$, $U$, and $\Gamma_0$, corresponding to the energy of the impurity state, the onsite Coulomb repulsion, and by the scattering rate or exchange between the impurity and the con-duction electrons, respectively (Supplementary Note 7). In the asymmetric AIM, which is relevant to screening of vacancy magnetic moments in graphene, the particle-hole symmetry is broken by next-nearest neighbor hopping and by $U \neq 2|\varepsilon_d|$. The NRG phase diagram for this model is controlled by the valence fluctuation (VF) critical point, $\Gamma_C$[6–8,39,40]. At charge neutrality ($\mu = 0$), $\Gamma_C$ separates the NRG flow into two sectors: supercritical, $\Gamma_0 > \Gamma_C$, which flows to the asymmetric strong-coupling (ASC) fixed point where charge fluctuations give rise to a frozen impurity (FI) ground state[41], and subcritical, $\Gamma_0 < \Gamma_C$, which flows to the local moment (LM) fixed point where the impurity moment is unscreened. At the FI fixed point, the correlated ground state acquires one additional charge due to the enhancement of the particle-hole asymmetry in the RG flow. In a simplified picture, the fixed point spectrum can be understood by the flow of $\varepsilon_d \to -\infty$, leading to an effective doubly occupied singlet impurity state that decouples from the remaining conduction band[6,8,41]. In terms of the real physical orbitals, however, the NRG reveals a distribution of this additional charge between the

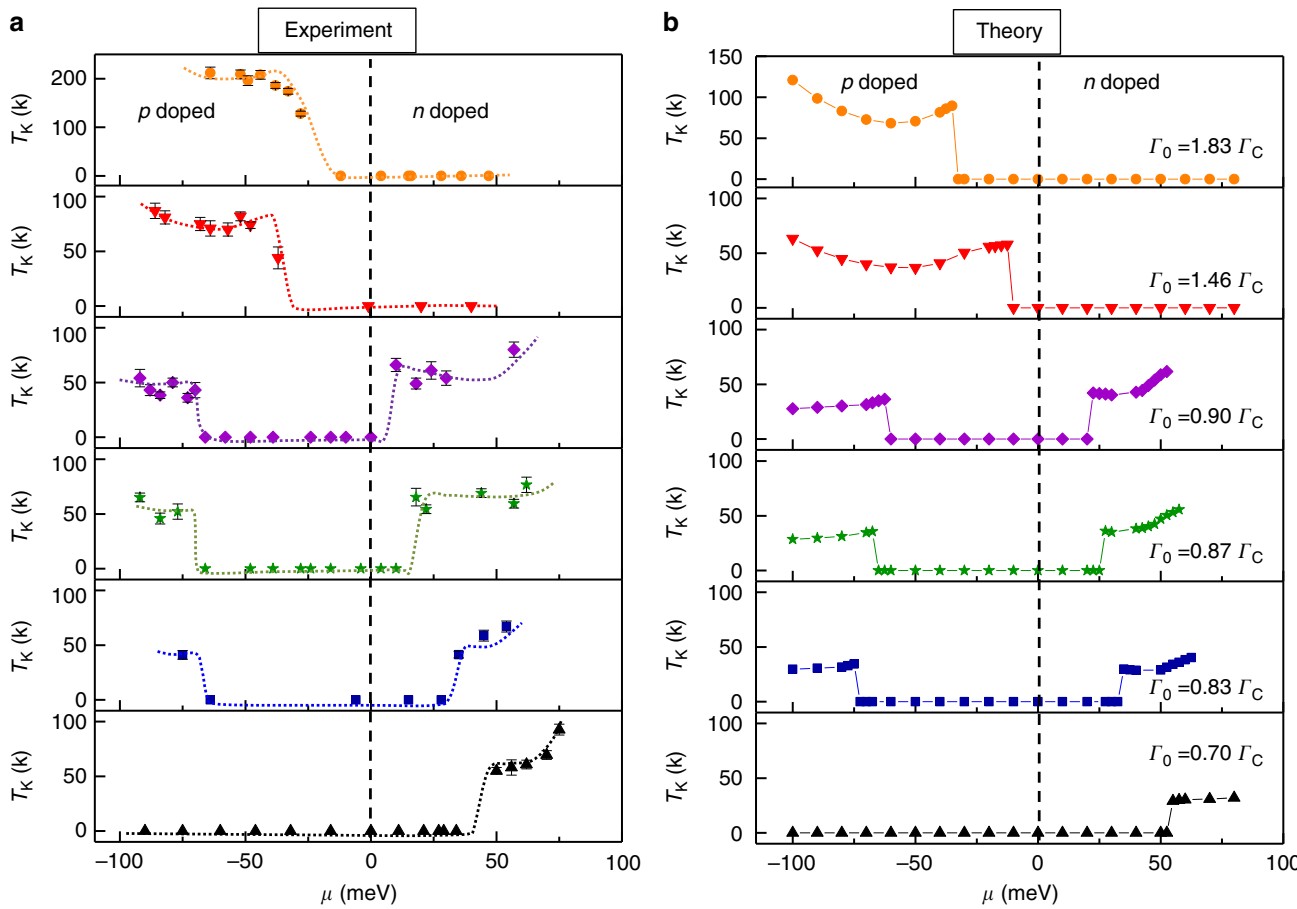

**Fig. 3** Chemical-potential dependence of the Kondo temperature. **a** Chemical potential dependence of $T_K$ obtained from the Fano lineshape fit of the Kondo peak. In the regions where the peak is absent we designated $T_K = 0$. **b** NRG result for the vacancies in panel **a**. $T_K$ is estimated by fitting the numerically simulated Kondo peak (Supplementary Note 3)

conduction band and the local orbital with a small enhancement of $n_\sigma = 1.2$–$1.3$. For $\Gamma_0 < \Gamma_C$ and $\mu \neq 0$, the appearance of relevant spin fluctuations gives rise to a cloud of spin-polarized electrons that screen the local moment below a characteristic temperature $T_K$ which is exponentially suppressed[8] ($\ln T_K \propto -1/|\mu|$). As a result, at sufficiently low doping, $T_K$ must fall below any experimentally accessible temperature, so that for all practical purposes its value can be set to zero (Fig. 4a). Using NRG to simulate the experimental spectra (Supplementary Note 7) we found $\varepsilon_d = -1.6$ eV for the bare σ-orbital energy[11,36], $U = 2$ eV[11,42,43] and a critical coupling $\Gamma_C = 1.15$ eV that separates the LM and the FI phases at $\mu = 0$. From the NRG fits of the STS spectra we obtained the value of the reduced coupling $\Gamma_0/\Gamma_C$ for each vacancy shown in Fig. 3 (Supplementary Note 8). The values, $\Gamma_0/\Gamma_C = 0.90$, and 1.83 obtained for the spectra in Fig. 2a, b place these two vacancies in the sub-critical and super-critical regimes, respectively.

In Fig. 3 we compare the chemical-potential dependence of the measured $T_K$, with the NRG results. The $T_K$ values are obtained from Fano-fits of the Kondo peaks leading to the $T_K$ ($\mu$) curves, shown in Fig. 3a. The corresponding values of $\Gamma_0/\Gamma_C$ and the $T_K$ ($\mu$) curves obtained by using NRG to simulate the spectra are shown in Fig. 3b. The close agreement between experiment and simulations confirms the validity of the asymmetric AIM for describing screening of vacancy spins in graphene. In Fig. 4a we summarize the numerical results in a $\mu$–$\Gamma_0$ phase diagram. At charge neutrality (defined by the $\mu = 0$ line), the critical point $\Gamma_0/\Gamma_C = 1$ signals a quantum phase transition between the LM phase and the FI phase[36]. The Kondo-screened phase appears at

finite doping ($\mu \neq 0$) and is marked by the appearance of the Kondo-peak,[8,9]. The phase diagram clearly shows the strong electron-hole asymmetry consistent with the asymmetric screening expected in in this system[4].

**Dependence of Kondo screening on corrugation amplitude.** Theoretical work[16,44] suggests that coupling of the vacancy moment with the conduction electrons in graphene may occur if local corrugations produce an out of plane component of the dangling σ-orbital. This removes the orthogonality restriction[17] that prevents hybridization of the σ-bands and π-bands in flat graphene and produces a finite coupling strength which increases monotonically with the out of plane projection of the orbital[18,19,45]. To check this conjecture we repeated the experiments for samples on substrates with different average corrugation amplitudes as shown in Fig. 4b, c. For consistency all the fabrication steps were identical. In the G/SiO$_2$ sample (single layer graphene on SiO$_2$) where the corrugation amplitude was largest (~1 nm), 60% of the vacancies displayed the Kondo peak and $T_K$ attained values as high as 180 K (Supplementary Note 5). For the flatter G/G/SiO$_2$ where the average corrugation was ~0.5 nm, we found that 30% of the vacancies showed the Kondo peak with $T_K$ values up to 70 K. For samples deposited on hBN, which were the flattest with local corrugation amplitudes of ~0.1 nm, none of the vacancies showed the Kondo peak. This is illustrated in Fig. 1e showing a typical $dI/dV$ curve on a vacancy in G/G/BN (double layer graphene on hBN) where a gate voltage of $V_g = -30$V was applied to separate the energies of $E_F$ and the DP. While this

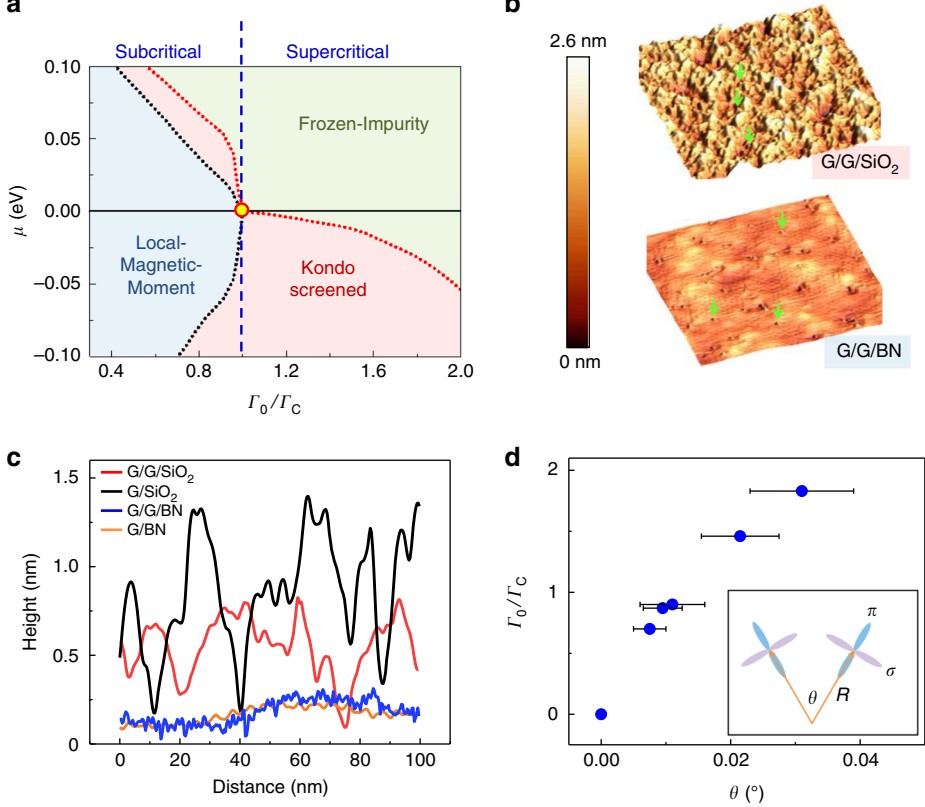

**Fig. 4** Quantum phase transition and Kondo screening. **a** $\mu - \Gamma_0$ phase diagram at 4.2 K. The critical coupling $\Gamma_C$ (circle at $\Gamma_0/\Gamma_C = 1.0$) designates the boundary between Frozen-Impurity and the Local-Magnetic-Moment phases at $\mu = 0$. Dotted lines represent boundaries between the phases (Supplementary Note 8). **b** STM topography for the G/G/SiO$_2$ (top) and G/G/BN (bottom) samples with the same scale bar ($V_b = -300$mV, $I = 20$pA). The arrows point to the vacancies. **c** Typical line profile of the STM topographies of graphene on different substrates with the same scanning parameters as in **b**. **d** The evolution of the hybridization strength with the curvature. Error bars represent the uncertainty in obtaining the angle between the σ-orbital and the local graphene plane orientation from the local topography measurements. Inset: sketch of the curvature effect on the orbital hybridization

spectrum shows a clear ZM peak, the Kondo peak is absent over the entire range of doping[28]. The absence of the Kondo peak in all the samples deposited on hBN highlights the importance of the local curvature. In order to quantify the effect of the local curvature on the coupling strength, we employed STM topography to measure the local radius of curvature, $R$, at the vacancy sites (Fig. 4d inset) from which we estimate the angle between the σ-orbital and the local graphene plane orientation[45], $\theta \approx a/2R$, where $a$ is the lattice spacing. We find that the coupling strength, $\Gamma_0/\Gamma_C$ ($\theta$), shows a monotonic increase with $\theta$ (Fig. 4d), consistent with the theoretical expectations[16,18,45]. Interestingly, the effect of the curvature on the Kondo coupling was also observed for Co atoms deposited on corrugated graphene[46], and was also utilized to enhance the spin-orbit coupling[47].

## Discussion

The results presented here shed light on the contradictory conclusions drawn from earlier magnetometry[20,21] and transport measurements[22] on irradiation induced vacancies in graphene. While the transport measurements revealed a resistivity minimum and logarithmic scaling indicative of Kondo screening with unusually large values of $T_K \sim 90$ K, magnetometry measurements showed Curie behavior with no evidence of low-temperature saturation, suggesting that the vacancy moments remained unscreened. To understand the origin of this discrepancy we note that magnetometry and transport are sensitive to complementary aspects of the Kondo effect. The former probes the magnetic moment and therefore only sees vacancies that are not screened,

while the latter probes the enhanced scattering from the Kondo cloud which selects only the vacancies whose moment is Kondo screened. Importantly, these techniques take a global average over all the vacancies in the sample. This does not pose a problem when all the impurities have identical coupling strengths. But if there is a distribution of couplings ranging from zero to finite values, as is the case here, global magnetization and transport measurements will necessarily lead to opposite conclusions as reported in the earlier work.

The local spectroscopy technique employed here made it possible to disentangle the physics of Kondo screening in the presence of a distribution of coupling strengths. This work demonstrates the existence of Kondo screening in a pseudogap system and identifies the quantum phase transition between a screened and an unscreened local magnetic moment. It further shows that the local magnetic moment can be tuned both electrically and mechanically, by using a gate voltage and a local curvature, respectively.

## Methods

**Sample fabrication.** The G/G/SiO$_2$ samples consisted of two stacked graphene layers deposited on a 300 nm SiO$_2$ dielectric layer capping a highly doped Si chip (acting as the backgate electrode)[23,48,49]. The bottom graphene layer was exfoliated onto the SiO$_2$ surface and the second layer was stacked on top by a dry transfer process. PMMA and PVA thin films were used as the carrier in the dry transfer process. Au/Ti electrodes were added by standard SEM lithography, followed by a metal thermal deposition process. After liftoff, the sample was annealed in forming gas (H$_2$: Ar, 1:9) at 300 °C for 3 h to remove the PMMA residue, and further annealed overnight at 230 °C in UHV[23]. All other samples (G/SiO$_2$, G/G/BN, and G/BN) were fabricated by a similar layer-by-layer dry transfer process. To

introduce single vacancies in the graphene lattice, the device was exposed under UHV conditions to a beam of He$^+$ ions with energy 100 eV for 5 to 10 s, and further annealed at high temperature in situ[28].

**Scanning tunneling microscopy experiment**. Except where mentioned all the STM experiments were performed at 4.2 K. $dI/dV$ curves were collected by the standard lock-in technique, with 0.5 mV AC modulation at 473 Hz added to the DC sample bias[2,24,50]. The chemical potential was tuned by the backgate voltage as illustrated in Fig. 1b.

**Data availability**. The data that support the findings of this study are available in Supplementary Information and from the corresponding authors upon reasonable request.

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

## Acknowledgements

We acknowledge support from DOE-FG02-99ER45742 (E.Y.A. and J.M.), NSF DMR 1708158 (Y.J.), Ministry of Science and Technology and also Academia Sinica of Taiwan (G.Y.G. and P.W.L.), Deutsche Forschungsgemeinschaft via project AN-275/8-1 (D.M. and F.B.A.), Key Research Program of the Chinese Academy of Sciences XDPB08-1 (J.M.). We thank Natan Andrei, Bruno Uchoa and Mohammad Sherafati for useful discussions.

## Author contributions

Y.J., J.M., and E.Y.A. conceived the work and designed the research strategy. Y.J. and J.M. performed the experiments, analyzed data, and wrote the paper. G.L. built the STM. P.W.L., D.M., G.Y.G., and F.B.A. carried out the theoretical work. T.T. and K.W.

contributed the boron nitride. E.Y.A. directed the project, analyzed the data, and wrote the paper.

## Additional information

**Competing interests:** The authors declare no competing interests.

