## [Peer Review File · Nature Communications]

Reviewers' comments:

Reviewer #1 (Remarks to the Author):

The authors study the Kondo screening of magnetic vacancies in graphene, using a combination of Scanning Tunneling Microscopy (STM) measurements and Numerical Renormalization Group (NRG) simulations. The impurities are identified at the surface of the sample by monitoring jointly the topographic spatial structures and the spectral signatures (Kondo anomalies). Both signals strengthen clearly the analysis.

An extensive study is then offered with respect to both chemical potential (tuned by a gate) and hybridization (tuned by the local environment of several impurities), showing drastic changes in the screening properties. This manuscript is certainly interesting to the field of quantum impurities, and well benchmarked with theoretical models. It may be suitable for publication in Nature Communications.

However, I would like the authors to clarify several points. My main criticism concerns the role of the chemical potential and the fate of the local moments. Naively I would have thought that as soon as the chemical potential is off the Dirac point, screening will necessarily occur (of course the resulting Kondo temperature may be exponentially small, and thus below the experimental temperature range). However, the author draw a phase diagram (in agreement with the NRG calculations that they show) a large portion of which presents unscreened moments, with abrupt first-order-like transitions to the screened phase. I have a hard time to understand this aspect of the physics. In some references found in the bibliography (e.g. Ref [5]), a similar phase diagram is discussed, but the vertical axis is the local potential scattering on the impurity, which is certainly not equivalent to a global chemical potential acting on the electron gas. This central question should be clarified by the authors.

Related to the above issue, I fail to see why the frozen impurity regime (which seems to trivially result from fully occupying the level) does touch the quantum critical point. This feature does not seem generic in the pseudogap Anderson impurity model, and may have been fine tuned in the theory to match the experimental data. Again, some clarification is certainly needed here.

Finally, the title and conclusion mention a "mechanical control" of atomic moments, and I found this statement confusing, because control means rather something that one can act deliberately upon (like a gate). I did not see where such a control is achieved, rather it seems that the experiment only picks at random impurities that have a

slightly different local environment. If so, I would suggest to the author to remove this concept in their presentation.

Reviewer #2 (Remarks to the Author):

The paper entitled "Electrically and Mechanically Controlled Atomic-Magnetism in Graphene" by Jiang et al. reports STM measurements and a theoretical analysis of vacancy induced magnetism and Kondo effect in graphene sheets. While the magnetic moment associated with the so-called zero mode of the vacancy cannot lead to Kondo effect because of ferromagnetic coupling to the π conduction band, it has been proposed in the literature that the local moment associated with the dangling sigma bond at the vacancy, may lead to Kondo effect in the presence of a local curvature. The curvature removes the strict orthogonality between the sigma orbital and the π conduction band of flat graphene.

In this work the Kondo effect at vacancies induced by local curvature is investigated systematically by STM spectroscopy supported by NRG calculations. The Kondo effect is identified via the characteristic Kondo-Fano lineshape in the STM spectra, which are pinned to the Fermi level as the chemical potential is changed, while the zero mode resonance moves with the Dirac point. Investigating the gate dependence of the Kondo effect, two types of vacancies are identified, ones with subcritical and ones with supercritical coupling to the conduction electrons. The Kondo temperature as a function of the gate voltage is extracted from the width of the measured Fano lineshapes for each vacancy. Comparison with NRG calculations of a single impurity pseudogap Anderson model, then allows to determine the relative coupling strengths for each vacancy. From the numerical results a phase diagram for the pseudogap Kondo effect as a function of the gate and relative coupling strength is obtained. Moreover, the frequency of occurrence of the Kondo effect at vacancies is linked to the corrugation of the graphene samples on different substrates, showing that the frequency increases with the average corrugation of the sample.

An important conclusion of the paper is that the seemingly contradictory experimental results of transport and magnetometry measurements of irradiation induced vacancies in graphene, are due to the distribution of couplings for different vacancies in the same graphene sheet leading also to a distribution of behaviors with respect to Kondo screening. The complementary character of transport and magnetometry measurements then necessarily leads to opposite conclusions.

I think that this is a very nice paper. It is very well written and easy to follow. The experimental results and their analysis are substantial and presented in a systematic way. The theoretical analysis in terms of the pseudogap Anderson model and its numerical solution nicely complements the experimental results. Overall the paper gives a clear picture of the rather complex situation of vacancy induced magnetism and Kondo effect in graphene. Therefore I can recommend publication in Nature Communications. I only a few minor comments that the authors should consider:

1) I was briefly confused by the choice of model to explain the experimental results. Since the dangling sigma orbital couples ferromagnetically to the zero mode moment, I first thought that a two-orbital Anderson model would be necessary to properly describe the situation, expecting some kind of underscreened Kondo physics in addition to the pseudo gap Kondo effect. Of course after a while it dawned on me that due to the sharpness of the ZM resonance, it would only carry a moment close to charge neutrality where the Kondo effect is suppressed anyway. So maybe one or two sentences in this direction at the beginning of the theoretical paragraph to justify the choice of the single impurity Anderson model would help to avoid such confusion. On the other hand, a little bit of wracking one's brains from time to time is also quite healthy.

2) It would be nice if the occupancy of the impurity level (as a function of gate and Γ_0) in

the NRG calculations was reported somewhere.

3) The choice of $U=2\text{eV}$ for the NRG calculations is justified by this being the LDA value in the Supplementary Information. I don't understand what is meant by this. The references cited in this context are also not very helpful in this regard. I guess the authors mean that this value was somehow extracted from a density functional theory (DFT) calculation of graphene with vacancies employing the local density approximation (LDA). The authors should be more specific and cite the proper reference. In any case, I think it is also fair to take U just as a fitting parameter.

----- **Reply to Reviewer 1** -----

We thank Reviewer 1 for valuable comments and for pointing out that “this manuscript is interesting to the field of quantum impurities, and well benchmarked with theoretical models.” We have considered all the comments/suggestions raised in the Reviewer’s report and revised our manuscript accordingly. Our point by point response is detailed below.

Reviewer 1 wrote:

My main criticism concerns the role of the chemical potential and the fate of the local moments. Naively I would have thought that as soon as the chemical potential is off the Dirac point, screening will necessarily occur (of course the resulting Kondo temperature may be exponentially small, and thus below the experimental temperature range). However, the author draw a phase diagram (in agreement with the NRG calculations that they show) a large portion of which presents unscreened moments, with abrupt first-order-like transitions to the screened phase. I have a hard time to understand this aspect of the physics.

Authors reply:

The reviewer is correct in pointing out that, strictly speaking, the impurity moment should be Kondo screened for any finite value of the chemical potential, μ . This question was addressed by Vojta, Fritz and Bulla in their 2010 article cited in reference 6. These authors used weak-coupling RG to show that for sub-critical coupling strengths at finite μ the Kondo temperature is exponentially suppressed: $\ln T_K \propto -1/|\mu|$, as the reviewer also noted. As a result, at sufficiently low doping the value of T_K must fall below any experimentally accessible temperature, so that for all practical purposes its value can be set at 0. As shown in Figure 5 of reference 6, for the experimental parameters relevant to this work ($T=4\text{K}$ and $\Gamma < 0.9 \Gamma_c$), the crossover between screened and unscreened moments as a function of doping is very sharp, which allows us to label the different regimes as distinct phases. The phase diagram presented in Fig. 4A of the manuscript classifies the various regimes at the experimental temperature of 4 K using the above criterion. We have added a comment in the SI to further clarify this point.

Reviewer 1 wrote:

In some references found in the bibliography (e.g. Ref [5]), a similar phase diagram is discussed, but the vertical axis is the local potential scattering on the impurity, which is certainly not equivalent to a global chemical potential acting on the electron gas. This central question should be clarified by the authors.

Authors reply:

As the referee correctly points out, in Ref [5] Fritz and Vojta did not consider the effect of finite chemical potential and temperature on the phase diagram, but restricted their work to

the influence of the local PH symmetry breaking on the RG flow in the $T = 0$ phase diagram, which can be included in a local scattering term. It was not until 2010 that these authors (reference 6) extend their work to include the effect of finite temperature and finite chemical potential.

Reviewer 1 wrote:

Related to the above issue, I fail to see why the frozen impurity regime (which seems to trivially result from fully occupying the level) does touch the quantum critical point. This feature does not seem generic in the pseudogap Anderson impurity model, and may have been fine tuned in the theory to match the experimental data. Again, some clarification is certainly needed here.

Authors reply:

We thank the reviewer for this question, which is far from trivial. The presence of the frozen impurity regime and the fact that it generically touches the critical point at charge neutrality was first shown by Gonzalez-Buxton and Ingersent (1998 Phys. Rev. B 57 14254) in the context of NRG calculations for a pseudogap asymmetric Anderson impurity model coupled to a fermionic bath with a linear DOS. This work was later extended by Fritz and Vojta (Vojta M and Fritz L 2004 Phys. Rev. B 70, 094502; idem 70, 214427) to the case of graphene as a function of doping. At charge neutrality Fritz and Vojta showed the existence of a so called valence fluctuation critical point (which corresponds to our Γ_c) that separates the NRG flow to the LM fixed point ($\Gamma_0 < \Gamma_c$ corresponding to the LM phase) from the flow to the ASC fixed point ($\Gamma_0 > \Gamma_c$ corresponding to the FI phase).

If the impurity is completely decoupled from the conduction band then, as the referee correctly points out, the FI would trivially imply double occupancy. In the presence of finite coupling however, a more subtle picture emerges: the local impurity, whose occupation is between single and double, is surrounded by a cloud of conduction band charge fluctuations. The resulting entangled FI state, comprising the impurity and the cloud of charge fluctuations, is a doubly occupied singlet state where spin fluctuations are frozen.

Following the reviewer's suggestion we have included a detailed discussion of the NRG phase diagram and of the FI state in the main text. A discussion of the standard Kondo model and its comparison to the asymmetric Anderson impurity model was added in a new section, SI 9.

Reviewer 1 wrote:

Finally, the title and conclusion mention a "mechanical control" of atomic moments, and I found this statement confusing, because control means rather something that one

can act deliberately upon (like a gate). I did not see where such a control is achieved, rather it seems that the experiment only picks at random impurities that have a slightly different local environment. If so, I would suggest to the author to remove this concept in their presentation.

Authors reply:

Following the reviewer's suggestion we have changed the title to "Inducing Kondo screening in graphene with gating and local-curvature"

Furthermore, we have replaced "the ability to electrostatically control" with "the ability to electrostatically tune" in the Abstract, and "local magnetic moment can be controlled" with "local magnetic moment can be tuned" in the last paragraph of the revised manuscript.

----- **Reply to Reviewer 2** -----

We thank Reviewer 2 for valuable comments and for the positive remarks: “this is a very nice paper. It is very well written and easy to follow. The experimental results and their analysis are substantial and presented in a systematic way. The theoretical analysis in terms of the pseudogap Anderson model and its numerical solution nicely complements the experimental results. Overall the paper gives a clear picture of the rather complex situation of vacancy induced magnetism and Kondo effect in graphene. Therefore I can recommend publication in Nature Communications.” We have considered all the comments/suggestions raised in the Reviewer’s report and revised our manuscript accordingly. Our point by point response is detailed below.

Reviewer 2 wrote:

1) I was briefly confused by the choice of model to explain the experimental results. Since the dangling sigma orbital couples ferromagnetically to the zero mode moment, I first thought that a two-orbital Anderson model would be necessary to properly describe the situation, expecting some kind of underscreened Kondo physics in addition to the pseudo gap Kondo effect. Of course after a while it dawned on me that due to the sharpness of the ZM resonance, it would only carry a moment close to charge neutrality where the Kondo effect is suppressed anyway. So maybe one or two sentences in this direction at the beginning of the theoretical paragraph to justify the choice of the single impurity Anderson model would help to avoid such confusion. On the other hand, a little bit of wracking one's brains from time to time is also quite healthy.

Authors reply:

We are delighted that the referee’s “brain-wracking” bore fruit. However for the benefit of less diligent readers we have added a discussion to clarify this point at the beginning of the theoretical paragraph.

In addition, we have uploaded a related theoretical work on the cond-mat arXives (ref 36 in the main text) where the effect of the two orbitals was treated in detail.

Reviewer 2 wrote:

2) It would be nice if the occupancy of the impurity level (as a function of gate and Γ_0) in the NRG calculations was reported somewhere.

Authors reply:

In the FI state the occupation number of the bare impurity is intermediate between 1 and 2. However, since the FI is an entangled state comprising the bare impurity surrounded by a cloud of conduction band charge fluctuations, the occupation number of the bare impurity state in itself has no physical significance. On the other hand the entangled FI state is a doubly occupied spin singlet and its occupancy is independent of gate or coupling. Therefore, although it is quite straightforward to calculate the bare impurity occupancy in the FI state, we prefer not to include it in this work. We have revised the main text and SI to clarify this point. We would like to point out however that the calculations concerning the occupancy of the impurity level as a function of chemical potential and Γ_0 are included in the theoretical cond-mat manuscript mentioned above.

Reviewer 2 wrote:

The choice of $U=2\text{eV}$ for the NRG calculations is justified by this being the LDA value in the Supplementary Information. I don't understand what is meant by this. The references cited in this context are also not very helpful in this regard. I guess the authors mean that this value was somehow extracted from a density functional theory (DFT) calculation of graphene with vacancies employing the local density approximation (LDA). The authors should be more specific and cite the proper reference. In any case, I think it is also fair to take U just as a fitting parameter.

Authors reply:

We agree with the Reviewer that U is treated more or less as a fitting parameter here. Nonetheless, one should start a fitting process with reasonable values of U and ϵ_d . Therefore, based on our previous work [34] and also the doubly occupied σ -orbital energy deduced from previous LDA calculations (e.g., [39]), we started the fitting process with the educated guess of $U = 2 \text{ eV}$ and $\epsilon_d = -1.5 \text{ eV}$. To avoid unnecessary confusion, we have replaced the phrase “Starting with the LDA value $U = 2 \text{ eV}$ ” with “Starting with the initial value $U = 2 \text{ eV}$ ” in Sec. 8 in the revised Supplementary Materials.

REVIEWERS' COMMENTS:

Reviewer #1 (Remarks to the Author):

The authors have addressed carefully all the questions, and the manuscript has gained in clarity. I will thus recommend its publication in its present form. If possible, I'd like the authors to make a small modification to the main text. They have given a welcomed discussion in the Supplementary Material of the fact that the local moment "phase" is rather a regime where the Kondo temperature is exponentially smaller than the base temperature of the device. It would be good if this remark appeared (in short) in the main text and in the relevant figures. That would really help the readers who won't necessary dig into the Supplementary Material.

Reviewer #2 (Remarks to the Author):

I think the authors have satisfyingly replied to the comments of both referees, and the paper has been revised accordingly. As said, before, this is definitely a very nice and interesting paper. I therefore recommend publication in Nature Communications.

Response to Reviewer's comments:

Reviewer #1 (Remarks to the Author):

The authors have addressed carefully all the questions, and the manuscript has gained in clarity. I will thus recommend its publication in its present form. If possible, I'd like the authors to make a small modification to the main text. They have given a welcomed discussion in the Supplementary Material of the fact that the local moment "phase" is rather a regime where the Kondo temperature is exponentially smaller than the base temperature of the device. It would be good if this remark appeared (in short) in the main text and in the relevant figures. That would really help the readers who won't necessary dig into the Supplementary Material.

Answer: We thank the reviewer for this useful suggestion. We have added this point about the Local Moments phase in the caption to Figure 4a of the main text.

Reviewer #2 (Remarks to the Author):

I think the authors have satisfyingly replied to the comments of both referees, and the paper has been revised accordingly. As said, before, this is definitely a very nice and interesting paper. I therefore recommend publication in Nature Communications.

Answer: We thank the reviewer for appreciating our paper.